# New Opportunities in the Systemic Treatment of Hepatocellular Carcinoma—Today and Tomorrow

**DOI:** 10.3390/ijms25031456

**Published:** 2024-01-25

**Authors:** Rafał Becht, Kajetan Kiełbowski, Michał P. Wasilewicz

**Affiliations:** 1Department of Clinical Oncology, Chemotherapy and Cancer Immunotherapy, Pomeranian Medical University in Szczecin, 71-252 Szczecin, Poland; rafal.becht@pum.edu.pl (R.B.); kajetan.kielbowski@onet.pl (K.K.); 2Liver Unit, Department of Gastroenterology, Pomeranian Medical University in Szczecin, 71-252 Szczecin, Poland

**Keywords:** hepatocellular carcinoma, targeted therapy, immunotherapy, tyrosine kinase inhibitors, monoclonal antibodies

## Abstract

Hepatocellular carcinoma (HCC) is the most common primary liver cancer. Liver cirrhosis, hepatitis B, hepatitis C, and non-alcoholic fatty liver disease represent major risk factors of HCC. Multiple different treatment options are available, depending on the Barcelona Clinic Liver Cancer (BCLC) algorithm. Systemic treatment is reserved for certain patients in stages B and C, who will not benefit from regional treatment methods. In the last fifteen years, the arsenal of available therapeutics has largely expanded, which improved treatment outcomes. Nevertheless, not all patients respond to these agents and novel combinations and drugs are needed. In this review, we aim to summarize the pathway of trials investigating the safety and efficacy of targeted therapeutics and immunotherapies since the introduction of sorafenib. Furthermore, we discuss the current evidence regarding resistance mechanisms and potential novel targets in the treatment of advanced HCC.

## 1. Introduction

Primary liver cancer is the third leading cause of cancer-related deaths, following lung and colorectal cancers. Hepatocellular carcinoma (HCC) is the most common primary liver cancer representing approximately 75% to 85% of cases [1]. Liver cirrhosis represents an important risk factor for HCC development. Furthermore, HBV and HCV infections, as well as non-alcoholic fatty liver disease (NAFLD) can cause HCC [2]. Multiple treatment methods are available for patients with HCC, including ablation, surgical resection, liver transplantation, transarterial chemoembolization (TACE), or systemic therapy. Treatment strategy depends on the stage of the disease. Barcelona Clinic Liver Cancer (BCLC) classifies HCC into five categories (0, A, B, C, and D), which are based on the number and size of lesions, performance status, liver function, vascular invasion, and extrahepatic spread, as well as tumor burden. Systemic treatment is recommended for a cohort of patients with BCLC stage B, who will not benefit from TACE (infiltrative, diffuse, and extensive bilobar involvement), and for patients with BCLC stage C [3]. During the last 15 years, systemic treatment for patients with advanced HCC has changed enormously and novel targeted agents and immunotherapies are now approved and recommended in the treatment guidelines. The aim of this review is to discuss the major clinical trials that led to the approval of current therapies, as well as to present potential future combinations and targets that might be useful in the treatment of HCC.

## 2. Targeted Therapy in Hepatocellular Carcinoma

### 2.1. Receptor Tyrosine Kinases in Hepatocellular Carcinoma

Receptor tyrosine kinases (RTKs) are key regulators of major cellular processes, including differentiation, proliferation, and survival, among others. The binding of growth factors to their respective RTKs recruits downstream enzymes and activates signaling pathways, such as PI3K/AKT, MAPK, and JAK/STAT. Abnormal activation of RTKs and their downstream elements may lead to constant proliferation and resistance to apoptosis, which is a hallmark of tumorigenesis. Autocrine activation, amplification, mutations, or chromosomal rearrangements are the main reasons for constitutively activated RTKs, which, in turn, drive tumor progression [4]. Throughout the years, multiple studies have demonstrated that several RTKs are implicated in the pathogenesis of HCC. These pathways may be inhibited by a targeted therapy using monoclonal antibodies or tyrosine kinase inhibitors (TKIs). The latter may be a selective drug that inhibits a single molecule or multi-TKI that targets several RTKs. 

To begin with, one of the important mechanisms regulated by RTKs and their ligands is angiogenesis. Neovascularization is a typical process for tumorigenesis, as new vessels are the sources of oxygen and nutrients for proliferating malignant cells. Vascular endothelial growth factor (VEGF), fibroblast growth factor (FGF), angiopoietins (Ang), platelet-derived growth factors (PDGF), and epidermal growth factor (EGF) are some of the growth factors involved in angiogenesis [5,6,7]. Importantly, several studies have demonstrated that angiogenesis-related growth factors/RTKs are expressed by the HCC cells, and their knockdown or blockage suppresses the viability of these cells. For instance, an early study by von Marschall et al. demonstrated that VEGF is strongly expressed by HCC cells compared with hepatocytes from cirrhotic or normal liver [8]. Moreover, the VEGF receptor (VEGFR) has also been found to be expressed by HCC cells. Most importantly, VEGF plays a role in HCC migration, survival, or progression from non-alcoholic fatty liver disease [9,10]. It also regulates tumor immunity, by augmenting PGE2- and IL-10-induced FasL expression on endothelial cells, which kills effector T cells [11]. Furthermore, mRNA expression of PDGF, FGF, and Ang has also been found elevated in cancer tissues [12,13,14]. FGF8, one of the members of the FGF family, stimulates the proliferation of HCC cells [15]. Moreover, forced overexpression of EGF enhances the development of HCC lesions in transgenic mice [16]. Therefore, despite being involved in physiological angiogenesis, many growth factors are upregulated in HCC tissues and can have a direct impact on tumor progression. As a result, targeting RTKs or their ligands has been suggested as a possible therapy in advanced and unresectable HCC. 

### 2.2. Sorafenib

Sorafenib is a multi-target agent that inhibits Raf-1, wild-type, and V599E mutant BRAF (members of the Ras/Raf/Mek/Erk pathway), as well as VEGFR, PDGFR, Fms-like tyrosine kinase (FLT-3), and, hepatocyte factor receptor (c-Kit), among others [17,18]. Moreover, recent studies have shown that sorafenib has immunomodulatory properties. For instance, it promotes NK cell proliferation and the production of cytotoxic molecules [19]. Sorafenib was approved by the Food and Drug Administration (FDA) in 2007 for the treatment of unresectable HCC [20]. An early phase 3 double-blind clinical trial (SHARP, NCT00105443) investigated the efficacy of sorafenib compared to placebo in patients with advanced HCC without prior systemic treatment. Overall survival (OS) was higher in the sorafenib arm (10.7 vs. 7.9 months, respectively). In addition, patients in the sorafenib group had a longer time to radiologic progression (5.5 vs. 2.8 months, respectively) [21]. Another phase 3 double-blind placebo-controlled trial (AP, NCT00492752) with patients from the Asia-Pacific region showed superior survival in the sorafenib arm (6.5 vs. 4.2 months, respectively) [22]. A phase 2 trial (NCT00108953) also evaluated the efficacy of the first-line sorafenib, but combined with doxorubicin. Patients in the doxorubicin + sorafenib cohort achieved greater OS and PFS compared to the doxorubicin + placebo arm (13.7 vs. 6.5 months and 6 vs. 2.7 months, respectively) [23]. However, when doxorubicin with sorafenib was compared to sorafenib alone in a subsequent phase 3 trial, the results showed that the addition of doxorubicin to sorafenib did not prolong OS or PFS, but increased the risks of grade 3 or 4 neutropenia and thrombocytopenia (36.8% vs. 0.6% and 17.5% vs. 2.4%, respectively) [24]. Subsequently, the sorafenib plus doxorubicin strategy has been investigated as a second-line treatment in patients with advanced HCC who progressed on sorafenib. Seventy-six percent of patients survived for 6 months, and the median OS and PFS were 8.6 and 3.9 months, respectively. As a result, the authors did not recommend this strategy in patients with HCC [25].

Sorafenib alone did not offer high response rates and further studies evaluated combinations of this multi-TKI with other therapeutic methods or agents. For instance, a phase 3 trial (NCT01214343; SILIUS) by Kudo and colleagues investigated the combination of sorafenib with low-dose cisplatin and fluorouracil hepatic arterial infusion chemotherapy (HAIC) in patients with advanced HCC and compared with the sorafenib monotherapy cohort. Median OS was similar (11.8 vs. 11.5 months) but the overall response was higher in the combination group compared to the monotherapy arm (36% vs. 18%, respectively) [26]. A similar response rate (34%) was observed in a recent single-arm study, which evaluated the efficacy of sorafenib with intermittent cisplatin HAIC [27]. Moreover, in a randomized phase 2 study of oral sorafenib plus oxaliplatin and 5-fluorouracil HAIC vs. sorafenib alone in patients with inoperable advanced HCC with major portal vein tumor thrombosis (NCT03009461), Zheng et al. demonstrated that higher OS, PFS, and ORR were noted in the former group (16.3 vs. 6.5 months; 9 vs. 2.5 months; 41 vs. 3%, respectively) [28]. A combination of sorafenib with bavituximab, a monoclonal antibody that targets phosphatidylserine, has been recently evaluated in a single-arm phase 2 trial (NCT01264705). The median OS, PFS, and ORR were 6.1 months, 4.8 months, and 5%, respectively [29]. Interestingly, the addition of oral vitamin K to sorafenib treatment resulted in an increased response rate compared to sorafenib monotherapy (27.3% vs. 4.5%, respectively) [30].

Consequently, clinical trials showed that sorafenib offers improved efficacy compared to placebo. However, the efficacy of this multi-TKI may depend on the liver function. Importantly, the majority (97%) of included patients in the landmark SHARP and AP trials had preserved liver function (Child–Pugh A) [21,22]. Ten years after the SHARP trial, Labeur et al. demonstrated that clinical efficacy is reproducible for a similar population. In contrast, patients non-eligible for the SHARP trial criteria showed reduced OS (9.5 vs. 5.4 months, respectively). Patients who did not meet SHARP criteria had higher liver enzyme levels and had more often Child–Pugh B status. As a result, the authors suggested that sorafenib should be restricted for patients with Child–Pugh A status [31]. Moreover, the ALICE-1 study suggested that genetics may play an important role in the response to sorafenib, as certain genotypes within VEGF polymorphisms are associated with a significantly higher OS [32]. Despite unsatisfactory responsive rates observed in clinical trials, rare case reports describe patients who achieved a significant tumor size reduction, in whom surgical approach became feasible [33,34]. Nevertheless, in a large retrospective analysis by Rimola et al., out of 1119 HCC patients treated with sorafenib, only 12 (1%) were finally classified as complete responders. Importantly, the median duration of treatment and the OS of these patients were 40.1 and 85.8 months, respectively [35]. Importantly, several prognostic factors for the response during sorafenib treatment have been identified. For instance, based on the SHARP and AP trials, the presence of macrovascular invasion, tumor burden, high AFP, and bilirubin concentrations, as well as ECOG PS 1 or 2 (vs. 0), are some of the prognostic factors for poorer OS in multivariate analysis. In contrast, HCV-positive status, absence of extrahepatic spread, and low neutrophil-to-lymphocyte ratio were associated with greater benefits from sorafenib treatment [36]. Furthermore, the occurrence of dermatologic AEs has been associated with a higher probability of prolonged survival in patients treated with sorafenib [37]. An approval of targeted sorafenib represents a breakthrough in the treatment of unresectable HCC. Subsequently, several multi-target TKIs were evaluated as potential first-line agents in the treatment of advanced HCC. However, they did not show improved outcomes compared to sorafenib [38,39,40]. Another challenge was to find safe and efficient second-line therapeutics. 

### 2.3. Regorafenib

Regorafenib is an oral multi-target agent that suppresses PDGFR-β, VEGFR 1–3, Tie-2, c-Kit, FGFR-1, Ret, RAF-1, BRAF, and p38, and is structurally similar to sorafenib [41]. The drug was approved by the FDA in 2017 for the treatment of HCC patients previously receiving sorafenib. An early phase 2 single-arm trial (NCT01003015) evaluated the use of regorafenib as a second-line treatment in patients with advanced or intermediate HCC who were previously treated with sorafenib. The median OS was 13.8 months, and a 6-month survival rate was 79%. Overall response and disease control rates were 3% and 79%, respectively [42]. A subsequent phase 3 trial (RESORCE, NCT01774344) compared the efficacy of post-sorafenib regorafenib with placebo in HCC patients. The former arm showed superior OS, PFS, and ORR compared to placebo (10.6 vs. 7.8 months, 3.1 vs. 1.5 months, 11% vs. 4%, respectively). In the regorafenib cohort, the most common grade 3 or 4 adverse events included hypertension (15%), hand–foot skin reaction (13%), increased bilirubin and AST levels (10% each), and fatigue, which occurred in 9% of patients [43]. Therefore, regorafenib became the recommended post-sorafenib second-line agent. Importantly, worse outcomes were observed in the group of patients with Child–Pugh B status. Kim et al. showed that in this cohort, regorafenib treatment resulted in a response rate of 5.1%, while median OS and PFS were 4.6 and 1.8 months, respectively. As a result, the authors concluded that regorafenib should not be given to patients with Child–Pugh B and albumin–bilirubin grade 3 status [44]. Furthermore, an albumin–bilirubin (ALBI) marker has been suggested to predict response. In a meta-analysis by Xu and collaborators, the authors determined that a low-grade ALBI was associated with better survival [45]. 

### 2.4. Cabozantinib

Cabozantinib is an oral multi-TKI targeting VEGFR 1–3, RET, MET, and AXL, among others [46]. It was approved by the FDA in 2019 for the treatment of HCC previously treated with sorafenib. A phase 3 CELESTIAL trial (NCT01908426) compared the treatment with cabazantinib and placebo in patients previously treated with sorafenib (with up to two previous systemic treatments). The results showed significantly prolonged median OS and PFS in the cabozantinib cohort (10.2 vs. 8.0 months, *p* = 0.005; 5.2 vs. 1.9 months, *p* < 0.001, respectively). In the cabozantinib arm, the most common grade 3 or 4 AEs included palmar–plantar erythrodysesthesia (17%) and hypertension (16%) [47]. A real-world study evaluating the treatment of cabozantinib in a second- and third-line settings in HCC patients with preserved liver function (Child–Pugh A status) showed that the median OS and PFS were 12.1 months and 5.1 months, respectively. Four percent of patients achieved PR, while SD was observed in 59.3% of cases. Macrovascular invasion, higher levels of AFP (>400 ng/mL), and ECOG-PS > 0 were associated with shorter survival [48]. Similar to the previously described agents, patients with Child–Pugh B status achieved a shorter median OS and experienced more AEs [49]. 

### 2.5. Lenvatinib

Lenvatinib is an oral multi-TKI, targeting VEGFR 1–3, PDGFRα, FGFR 1–4, RET, and KIT [50]. It was approved by the FDA in 2018 for the treatment of unresectable HCC in the first-line treatment. Lenvatinib displays immunomodulatory properties as the drug reduces the proportion of monocytes and macrophages, as well as promoting the CD 8+ T cell population [51]. Furthermore, the drug induces immunogenic cell death (ICD) [52]. A phase 3 clinical trial (NCT01761266) investigated the efficacy of lenvatinib vs. sorafenib in the first-line treatment of advanced HCC patients. The investigated agent demonstrated non-inferiority compared to sorafenib in terms of median OS (13.6 vs. 12.3 months, respectively, HR = 0.92 (95% CI 0.79–1.06). Nevertheless, Lenvatinib showed a significant improvement in the median PFS (7.4 vs. 3.7 months, respectively, HR = 0.66, *p* < 0.0001). Similarly, the response rate was higher in the study group. Interestingly, the study showed higher survival in the sorafenib group compared to the previous trials. The authors suggested it could be a result of a higher post-sorafenib treatment rate compared to other trials. The most common ≥ grade 3 AE was hypertension, which occurred in 23% of patients in the lenvatinib group, as compared to 14% in the sorafenib cohort [53]. A recent real-life study compared lenvatinib and sorafenib as the first treatment methods in advanced HCC. The authors showed that patients treated with lenvatinib had a reduction in risk of death by 48% compared to sorafenib. The respective median PFS were 9 months and 4.9 months in the lenvatinib and sorafenib groups. Moreover, lenvatinib offered a significantly increased response rate (29.4% vs. 2.8%, respectively) [54]. Another study evaluated the use of first-line lenvatinib in 1325 patients with intermediate or advanced HCC not eligible for locoregional or surgical therapies. The median OS and PFS were 16.1 months and 6.3 months, respectively, while the ORR was 38.5% [55]. Treatment response seems to significantly depend on liver function, as demonstrated in a study by Ogushi et al. The authors observed that the ORR in patients with Child–Pugh A status was 36.5%, in contrast to 16.3% of patients with Child–Pugh B status [56]. The efficacy and safety of lenvatinib compared to sorafenib as a first-line treatment were recently evaluated in a meta-analysis that included 3908 patients. The study demonstrated no significant difference between both agents in terms of OS (*p* = 0.09). Nevertheless, the exclusion of two trials due to a high heterogeneity resulted in a significant difference where lenvatinib showed statistically significant superior outcomes in PFS (HR = 0.63, *p* < 0.00001). Additionally, the response rate was significantly higher in the lenvatinib group (25.74% vs. 6.4%, *p* < 0.00001). Safety profiles were comparable between both cohorts [57]. 

Since lenvatinib commenced use as a first-line agent, few studies have evaluated the therapeutic strategy after progression on lenvatinib. Tomonari et al. reported an analysis of 13 patients treated with sorafenib after progression on lenvatinib. The observed response differed depending on the criteria system used. According to the mRECIST, two and seven patients achieved PR and SD, respectively. In contrast, no patients achieved PR and nine patients had SD according to RECIST [58]. The use of regorafenib in a similar strategy in a small cohort (n = 28) resulted in an ORR of 10.7%. In terms of OS, the use of regorafenib as a second line in patients previously treated with sorafenib and lenvatinib was evaluated. Sequential treatment of lenvatinib with regorafenib achieved better results (OS: 15.9 vs. 11.7 months, *p* = 0.045). There was no significant difference between PFS in both groups [59]. In a large analysis by Casadei-Gardini et al., the authors described 827 patients who progressed during lenvatinib treatment and had a follow-up of more than 2 months after the start of a subsequent treatment. These patients received the best supportive care, sorafenib, transarterial chemoembolization, immunotherapy, ramucirumab, or regorafenib. Importantly, calculating from the start of lenvatinib, patients treated sequentially with immunotherapy (n = 60) achieved an extraordinary median OS of 47 months [55]. 

### 2.6. Anlotinib

Anlotinib is a multi-TKI, which targets VEGFR 1–3, c-Kit, FGFR 1–3, and PDGFR-α [60]. In a preclinical study by He et al. anlotinib enhanced apoptosis and inhibited the proliferation of HCC cells by suppressing Akt and ERK phosphorylation [61]. ALTER-0802, a phase 2 trial, evaluated the use of anlotinib in patients with locally advanced or metastatic HCC. The patients were divided into two groups depending on the prior use of TKIs. In these cohorts, patients who were not treated with TKIs previously achieved a median OS of 12.8 months. In contrast, the median OS achieved by the patients from the second group was 18 months. Importantly, recalculating the median OS from the start of the previous targeted treatment showed a median OS of 26.7 months [62]. 

### 2.7. Axitinib

Axitinib is an oral small-molecule agent more selective than previously described TKIs. It targets an ATP binding site of VEGFR 1, 2, and 3 [63]. Two single-arm phase 2 trials have been published evaluating the use of axitinib in a second-line setting in advanced HCC patients [64,65]. One randomized phase 2 trial has been performed, which compared axitinib/best supportive care (BSC) with placebo/BSC. The study found that there was no significant difference between the two groups in terms of OS (12.7 vs. 9.7 months, HR = 0.907, *p* = 0.287). Therefore, the study did not meet its primary endpoint. Similarly, there was no statistical significance in terms of ORR (9.7% vs. 2.9%, *p* = 0.091). Nevertheless, the median PFS was significantly longer in the study group (3.6 vs. 1.9, HR = 0.618, *p* = 0.004) [66]. Interestingly, in a subsequent analysis, after the exclusion of patients intolerant to previous antiangiogenic therapy, benefits in OS have been observed in Japanese patients [67]. 

### 2.8. Donafenib

Donafenib is a small multi-TKI, which targets VGFR and PDGFR, as well as Raf kinase. It has been approved for the treatment of unresectable HCC in China in 2021 [68]. Importantly, a phase II–III clinical trial showed that donafenib achieved higher median OS in patients with unresectable or metastatic HCC compared to sorafenib (12.1 vs. 10.3 months, HR = 0.831, *p* = 0.0245). Moreover, the survival rate at 18 months was higher in the donafenib cohort (35.4% vs. 28.1%, *p* = 0.046). Median PFS did not reach a statistical significance (3.7 vs. 3.6 months, HR = 0.909, *p* = 0.057). The investigated drug also demonstrated a beneficial safety profile compared to sorafenib. Importantly, this was the first phase 3 study to show a longer OS in the investigated drug compared to sorafenib in the first-line settings [69]. As demonstrated by a recent case report, donafenib is capable of inducing a long-term PFS. In a report by Li and Zhu, the authors describe a patient with BCLC stage 3 HCC who had been receiving donafenib for 31 months. After progression, the patient underwent radical surgery [70]. Recently, a retrospective analysis compared the treatment with donafenib and lenvatinib in patients with HCC BCLC stages B and C. The study showed that the donafenib group achieved better OS and PFS. ORR was higher in the donafenib cohort as well, but the difference was not statistically significant (32% vs. 20%, respectively, *p* = 0.113) [71]. 

### 2.9. Apatinib

Apatinib (also known as rivoceranib) is another small-molecule inhibitor that targets VEGFR 2. However, the drug also shows efficacy against c-Kit, Ret, and c-src [72]. A phase 3 trial (NCT02329860, AHELP) investigated the use of apatinib vs. placebo in patients intolerant or reflactory to previous lines of chemotherapy or targeted therapy. Apatinib showed a significantly better median OS (8.7 vs. 6.8 months, HR = 0.785, *p* = 0.048) and PFS (4.5 vs. 1.9 months, HR = 0.471, *p* < 0.0001). Hand–foot syndrome, hypertension, and decreased platelets were the most common grade 3/4 AEs in the study group and developed in 18%, 28%, and 13%, respectively [73]. 

### 2.10. Ramucirumab

Overall, TKIs targeting multiple RTKs, as well as more selective agents, provided a significant efficacy in the systemic treatment of advanced HCC. In addition to TKIs, monoclonal antibodies other than immunotherapy are also used in the treatment of HCC. Ramucirumab is a monoclonal antibody targeting VEGFR 2 [74], which was approved by the FDA in 2019 for the treatment of HCC patients previously treated with sorafenib and with AFP levels ≥ 400 ng/mL. In 2015, a phase 3 REACH trial (ramucirumab vs. placebo as a second-line treatment) did not meet its primary endpoint of achieving a significantly higher OS. However, in a subgroup of patients with AFP concentrations ≥400 ng/mL, the median OS was significantly higher (7.8 vs. 4.2 months, respectively, HR = 0.67, *p* = 0.006) [75]. As a result, a subsequent REACH-2 trial investigated ramucirumab vs. placebo in patients previously treated with sorafenib with serum AFP levels ≥400 ng/mL. The study achieved its primary endpoint and proved that the median OS was longer in the ramucirumab cohort (8.5 vs. 7.3 months, respectively, HR = 0.71, *p* = 0.0199) [76]. The results of the REACH-2 trial were the basis for the FDA approval of ramucirumab. Figure 1 presents summaries of the mechanism of action of targeted therapeutics discussed above.

## 3. Immunotherapy in Hepatocellular Carcinoma

### 3.1. Immunotherapy—Overview

The human immune system is designed to balance activation and suppression. Consequently, its dysregulation may result in the development of autoimmunity or immunosuppression. In the pathological cancerous environment, malignant cells use the inhibitory molecules expressed by T cells to evade immunity and promote immunosuppression, thus inhibiting the potential cytotoxic T cell capabilities. 

Briefly, T cells express immune checkpoints that negatively regulate their functions. Programmed cell death 1 (PD1) and cytotoxic T lymphocyte antigen 4 (CTLA4) are the most frequently mentioned molecules. They exert their biological functions by intracellular signaling pathways or by competing for co-stimulatory ligands. PD1 binds to its ligands PD-L1 and PD-L2, which are highly expressed by cancer cells [77]. Importantly, the development of agents targeting the negative regulators of T cells was a revolution in the field of oncology and introduced new possibilities for a number of different malignancies. These agents include pembrolizumab, durvalumab, atezolizumab, nivolumab, ipilimumab, and tremelimumab, among others. Despite their innovative mechanisms of action and promising efficacy, treatment with these agents is associated with specific immune-related adverse events (irAEs) [78]. In HCC, several immunotherapeutics have been approved for the treatment of advanced disease. In the previous paragraphs, we have described immune checkpoint inhibitors in combination with targeted therapeutics. However, recent trials have also investigated the safety and efficacy of immunotherapy alone. 

### 3.2. PD-1/PD-L Inhibitors

Nivolumab is a monoclonal antibody targeting PD1, which has been examined in several cancers in the series of CheckMate trials. In HCC, its efficacy and safety were investigated in the single-arm CheckMate 040 and phase 3 CheckMate 459 studies. In the first trial, the objective response was observed in 20% of patients (dose expansion cohort). Median OS was not reached, while the median PFS was 4 months [79]. In the subsequent CheckMate 459 phase 3 trial, nivolumab was evaluated in the first-line settings. The study group achieved a median OS of 16.4 months, which was not significantly different from the group treated with sorafenib, a targeted therapeutic that has been a mainstay of first-line treatment for many years [80]. In a multicenter retrospective “real-world” study by Fessas and collaborators, nivolumab was introduced in 233 patients as a first-to-fourth treatment line therapeutic. The ORR was achieved in 22.4% of patients while median OS was 12.2 months. Interestingly, the authors demonstrated a significant survival benefit in patients who achieved a complete response (CR) and partial response (PR). Median OS in these cohorts was 30.6 and 18.7 months, respectively [81]. 

Importantly, biomarkers that could select patients who are likely to respond or progress on treatment are of great interest. In the case of nivolumab, PD-L1 expression greater than 1% was associated with improved survival [82]. Furthermore, impaired liver function (Child–Pugh B and C), as well as jaundice, albumin ≤3.5 g/L, elevated liver enzymes, CRP concentration, and neutrophil-to-leucocyte ratio correlated with worse outcomes. Additionally, the number and dimensions of liver lesions were also associated with OS and PFS. Interestingly, the presence of pulmonary lesions greater than 30 mm was associated with a greater response to nivolumab compared to patients with smaller lesions or without pulmonary diseases [83].

Pembrolizumab is another monoclonal-antibody-targeting PD1 molecule, which does not stimulate complement nor Fc receptors and does not induce cytotoxic responses. The drug has been investigated in multiple cancers in the KEYNOTE trials [84]. In HCC, KEYNOTE-224, KEYNOTE-240, and KEYNOTE-394 were performed. The first study was a non-randomized phase 2 trial, which investigated the use of pembrolizumab as a second-line agent. The trial included 104 patients who achieved median PFS and OS of 4.9 and 12.9 months, respectively. Moreover, CR and PR were observed in a total of 17% of patients. IrAEs could be observed in 14% of participants and included hypothyroidism, adrenal insufficiency, and thyroiditis, among others [85]. Based on the results of the KEYNOTE-224, pembrolizumab received accelerated approval from the FDA. Similar positive trends were observed after an additional follow-up of approximately 2.5 years [86]. In the subsequent phase 3 trial (KEYNOTE-240), pembrolizumab was compared to the placebo. The trial demonstrated significantly better ORR for the immunotherapeutic cohort (18,3% vs. 4.4%, respectively) [87]. Recently, an updated follow-up study of the KEYNOTE-240 was published, which confirmed the positive trends of pembrolizumab treatment [88]. KEYNOTE-394 was a phase 3 trial that evaluated the use of pembrolizumab in patients from Asia. The study showed improved efficacy in the group treated with PD1 antibody [89]. These studies demonstrated promising results regarding the use of pembrolizumab in previously treated cohorts. Nevertheless, the drug was also investigated in patients who were not previously treated. Verset et al. published the results of cohort 2 of the KEYNOTE-224 study. The best-observed response was PR, which occurred in eight patients (16%). Median OS and PFS were 17 and 4 months, respectively [90].

A few studies examined potential prognostic biomarkers. Firstly, elevated concentrations of TGF-β were associated with worse outcomes in HCC patients treated with pembrolizumab [91]. Furthermore, the genomic landscape may indicate the potential response. Hong and collaborators found that all patients who did not respond to pembrolizumab harbored CTNNB1 mutations. Furthermore, treatment responders had enriched genes associated with hypoxia, TCR signaling, MHC, and immune checkpoints. By contrast, upregulation of genes associated with extracellular matrix, as well as HCC- and liver-specific genes, were observed in non-responders [92]. The efficacy of pembrolizumab was compared to that of nivolumab in a recent study by Chen and colleagues. There were no statistical differences between both cohorts in terms of OS, PFS, and ORR [93].

Tislelizumab is a humanized monoclonal antibody that also binds to PD1. Its efficacy and safety in malignancies are being examined in the series of RATIONALE trials. In HCC, RATIONALE-208 was a non-randomized phase 2 study, which included patients previously treated with at least one prior line of systemic therapy. The ORR, median OS, and PFS were 13%, 13.2, and 2.7 months, respectively [94]. Due to promising results, tislelizumab was further examined in the phase 3 RATIONALE-301 trial. In this study, 342 patients received tislelizumab, and the outcomes were compared with patients treated with sorafenib (n = 332). The PD1 inhibitor showed non-inferiority to sorafenib in terms of OS. On the other hand, more patients treated with tislelizumab achieved an objective response (14.3% vs. 5.4%). The cohort with sorafenib achieved longer median PFS, but the 12-month PFS rates were similar. In terms of safety, tislelizumab showed a favorable profile. IrAEs occurred in 18.3% of patients, but these events led to treatment discontinuation in only 3.3% of cases [95]. Consequently, tislelizumab might become another agent used in the first-line settings. 

### 3.3. Combination of PD1/PD-L1 Inhibitors with Agents Targeting CTLA-4

Notwithstanding PD1/PD-L1, several drugs targeting CTLA-4 have been developed to enhance cytotoxic responses. Interestingly, agents from these families could be combined to further improve the treatment outcomes. For instance, a combination of nivolumab (anti-PD1) with ipilimumab (anti-CTLA4), as well as durvalumab (anti-PD-L1) with tremelimumab (anti-CTLA-4), has been investigated. In a study by Yau et al. the authors divided 148 patients previously treated with sorafenib into three groups with different dosing. The ORRs of these groups were 32%, 27%, and 29%, respectively. The highest median OS (22.8 months) was achieved in Arm A (nivo 1 mg/kg + ipi 3 mg/kg every 3 weeks). However, this strategy was also associated with higher rates of AEs and irAEs than other groups [96]. Interestingly, a recent study evaluated the efficacy of this combination in patients previously treated with other immune checkpoint inhibitors. Median OS and PFS were 9.2 and 2.9 months, respectively. The treatment was associated with irAEs, including pneumonitis and colitis/esophagitis (13% each), followed by autoimmune hepatitis (9%), rash (9%), arthritis (3%), and myocarditis (3%) [97]. Thus, these studies demonstrated that nivolumab and ipilimumab could provide a clinical benefit in patients resistant to sorafenib and other PD-(L)1 inhibitors. Another combination that has been investigated in HCC involves durvalumab with tremelimumab. After achieving promising outcomes in a phase 2 trial including patients with unresectable HCC, this combination was evaluated in the HIMALAYA trial [98]. According to the recently published results of this study, tremelimumab + durvalumab achieved a significantly higher median OS compared to sorafenib (16.43 vs. 13.77 months, respectively). Moreover, more patients in the study group achieved an objective response (20.1% vs. 5.1%). However, the difference in PFS between these groups was not significant. In terms of safety, both groups demonstrated similar rates of grade 3–4 AEs (50.5% vs. 52.4%). Nevertheless, irAEs requiring treatment with glucocorticosteroids occurred in 20.1% of patients treated with tremelimumab + durvalumab, as compared to 1.9% in the sorafenib cohort [99]. 

### 3.4. Chimeric Antigen Receptor (CAR) T Cells

The name CAR-T comes from a common antigen receptor frequently found on a specific type of cancer cell. In classical immunotherapy based on stimulation of T lymphocytes, there are numerous limitations that cause tumor cells to be able to escape the supervision of T lymphocytes, and thus we observe a lack of therapeutic effect after immunotherapy. In CAR-T, the HCC patient’s own lymphocytes undergo a genetic modification that targets their own tumor cells. This genetic modification involves the insertion, via a vector (a modified lentivirus or retrovirus), of a gene encoding such a receptor, which enables recognition of the cancer cells. Good experience with CAR-T has been observed for several years in the treatment of B-lymphocyte-derived non-Hodgkin’s lymphoma [100] and acute lymphoblastic leukemia [101]. The good results of CAR-T use in hematologic malignancies have prompted the search for a place for this therapy in solid tumors, including HCC. However, unlike in hematologic malignancies, the use of CAR-T cells in solid tumors is challenging due to tumor heterogeneity in terms of antigen expression, difficult access of CAR-T cells to the tumor, and tumor microenvironment (TME) resistance to CAR-T therapy [102,103]. 

An early study by Zhu et al. demonstrated that the expression of glypican 3 (GPC3) is greater in HCC samples compared to benign hepatic diseases [104]. Subsequently, researchers have demonstrated the promising efficacy of these cells in in vitro experiments, as well as in mouse models with HCC xenografts obtained from human tumors [105]. In 2017, Jiang and co-authors examined the use of anti-GPC3 CAR-T cells in mouse xenograft models, and confirmed that GPC-3-CAR T-cell therapy is a promising candidate for the treatment of HCC [106]. These basic preliminary studies have given rise to numerous early-phase clinical trials. Currently, among the ongoing 22 early-phase clinical trials using CAR-T for the treatment of HCC, with as many as 11 trials using GPC3 as a target [103]. Among other therapeutic targets, CAR-T targeting CD147, EpCam, anti-mucin 1, anti-c-MET/PD-L1, and EGFR are being investigated [107]. Interestingly, CAR-T cells could be further modified to increase their efficacy. For instance, CAR-T-secreting IL-7 and CCL19 demonstrated greater preclinical anti-cancer activity compared to standard GPC3-targeting therapy. Furthermore, they have shown an important efficacy in a few HCC patients [108]. Similarly, anti-GCP3 CAR-T cells expressing IL-21 and IL-15 demonstrated significant efficacy against HCC xenografts [109]. We still have to wait for the results of many studies on the use of CAR-T in the treatment of HCC, but this procedure should be taken as a therapeutic option in recurrent and refractory HCC tumors.

## 4. Combination of Targeted Agents with Immunotherapy

### 4.1. Combination of TKI with Immunotherapy

In the previous paragraphs, we discussed the efficacy of targeted agents and immunotherapies in unresectable HCC. Importantly, recent studies have started to show important clinical benefits of combinations of targeted therapeutics with immune checkpoint inhibitors. First, a combination of regorafenib with immunotherapy in second-line treatment has been found to improve median PFS and OS [110,111]. Specifically, Huang et al. analyzed the efficacy of regorafenib in combination with sentilimab (anti-PD1) in patients after the treatment with sorafenib or lenvatinib. The ORR was 24.1% in the combination cohort, as compared to 9.1% in patients treated with regorafenib alone. Furthermore, the addition of sintilimab was associated with prolonged median OS (13.4 vs. 9.9 months) and PFS (5.6 vs. 4 months) [110]. Second, the combination of cabozantinib with an anti-PD1 agent has shown a higher anti-tumor efficacy than single agents in a preclinical study [112]. Recently, a phase 3 COSMIC-312 (NCT03755791) trial investigated the combination of cabozantinib with atezolizumab vs. sorafenib in patients with advanced HCC without previous systemic treatment. Importantly, the study only included patients with Child–Pugh A status. The results showed that patients in the combination cohort achieved a significantly longer median PFS (6.8 vs. 4.2 months, HR = 0.63, *p* = 0.0012). Grade 3 or 4 palmar–plantar erythrodysesthesia syndrome, AST, and ALT increase occurred in 8%, 9%, and 8% of patients in the combination group vs. 8%, 3%, and 2% in the sorafenib cohort, respectively [113]. Furthermore, a combination of lenvatinib with immunotherapy has been recently studied. LEAP-002 study (NCT03713593) compared the safety and efficacy of lenvatinib with pembrolizumab to lenvatinib alone. The results did not reach pre-specified efficacy boundaries, and the authors concluded that these findings do not support modification of clinical practice. Nevertheless, the median OS and PFS for both groups were 21.2 vs. 19 months and 8.2 vs. 8.0 months, respectively. Therefore, patients in the first group achieved one of the longest media OS described in a phase 3 trial evaluating first-line treatments for advanced HCC [114]. Xu and colleagues reported a real-world efficacy of lenvatinib combined with various PD-1 inhibitors (nivolumab, pembrolizumab, camrelizumab, sintilimab, tislelizumab, and toripalimab) in the treatment of unresectable HCC. Median OS and PFS were 17.8 and 6.9 months, respectively, while the ORR was 19.6%. Child–Pugh B, ECOG-PS 1,2 (vs. 0), involved organs (≥3), and high tumor burden scores were associated with shorter OS in a multivariate analysis [115]. Recently, lenvatinib combined with immunotherapy (sintilimab, tislelizumab, pembrolizumab, and camrelizumab) has been compared to regorafenib as a post-sorafenib treatment strategy. The combination cohort demonstrated a trend toward improved outcomes and response rates, but the differences were not significant [116]. Moreover, a recent study evaluated the use of cadonlimab, a bispecific antibody simultaneously binding to CTLA-4 and PD1, in combination with lenvatinib as a first-line treatment in patients with advanced HCC. The study investigated two strategies, but the ORR was approximately 35% in both cohorts. Importantly, median PFS and OS of all included patients were 9.7 and 26.9 months, respectively [117]. 

The use of anlotinib with toripalimab (anti-PD-1) as a first-line treatment also achieved a high median OS (18.2 months) and PFS (11 months). Hand–foot syndrome, hypertension, and abnormal liver function were the three most common ≥grade 3 AEs (9.7%, 9.7%, and 6.5%, respectively) [118]. A similar median PFS (12.2 months) was achieved in a recent phase 2 KEEP-G04 trial, which evaluated anlotinib combined with sintilimab (anti-PD-1) [119]. Since TKIs with immunotherapy may improve clinical benefits in patients with advanced HCC, a similar strategy may be performed with axitinib in the future. Only the phase 1b clinical study has been published so far evaluating the combination of axitinib with avelumab [120].

Importantly, a combination of apatinib with camrelizumab (anti-PD-1; phase 2, RESCUE trial) showed high response rates compared to previous studies [121]. In a randomized phase 3 trial (CARES-310) apatinib (rivoracenib) combined with camrelizumab achieved a significantly higher median OS (22.1 vs. 15.2 months, respectively, HR = 0.62, *p* < 0.0001), PFS (5.6 vs. 3.7 months, respectively, HR = 0.52, *p* < 0.0001) and overall response (25% vs. 6%, respectively, *p* < 0.0001) compared to sorafenib in the first-line setting [122]. Importantly, it was the first phase 3 study to report improved OS and PFS in combined treatment (anti-PD-1 + small molecule TKI) vs. TKI for unresectable HCC in the first-line setting. In addition, the median OS was longer than in the previously mentioned primary results from the LEAP-002 trial. Interestingly, in a single-arm phase 2 study, a combination of apatinib with sintilimab and capecitabine showed a long median PFS (9 months) and high response (ORR 50%) [123]. 

### 4.2. Bevacizumab/Atezolizumab

Bevacizumab is another monoclonal antibody targeting the VEGF/VEGFR axis. It binds to the circulating VEGF-A isoforms and suppresses neovascularization processes [124]. In clinical settings, bevacizumab is combined with atezolizumab as this strategy has been found to promote intratumoral CD8+ T cell population in renal cell carcinoma [125]. The FDA approval was based on the IMbrave150 phase 3 trial, which compared bevacizumab/atezolizumab with sorafenib as a first-line treatment in patients with unresectable HCC. The study group achieved significantly better OS and PFS (HR = 0.58, *p* < 0.001; HR = 0.59, *p* < 0.001, respectively). Moreover, the combination cohort also achieved better ORR (RECIST 27.3% vs. 11.9%, respectively) [126]. Recently, an update on the efficacy and safety of the IMbrave150 trial was published. The median OS and PFS in the study group were 19.2 and 6.9 months vs. 13.4 and 4.3 months in the sorafenib group, respectively [127]. The clinical outcomes seem to depend on the AFP fluctuations post-treatment. Low-stable and sharp-falling AFP trajectories have been correlated with longer PFS and OS compared to the high-rising trajectory [128]. Recently, a panel of AFP, ALP, and eosinophil count, as well as baseline CD8+ and CD8 + PD-L1+ lymphocytes, was associated with OS and response in patients treated with atezo/bev, respectively [129,130]. Furthermore, clinical outcomes of treatment with atezo/bev depend on the line of treatment. First-line strategy is associated with longer PFS compared to the use of atezo/bev in later-line settings [131]. As a result, atezo/bev proved to be an efficient first-line treatment in patients with unresectable HCC. Nevertheless, a more recent study compared the efficacy of atezo/bev with lenvatinib, a multi-TKI that previously showed non-inferiority to sorafenib in the phase 3 trial. Casadei-Gardini and colleagues formed two cohorts of patients receiving atezo/bev (n = 864) and lenvatinib (n = 1343). The authors found no statistically significant difference in terms of OS between both groups. However, the use of lenvatinib could be more beneficial in patients with NASH/NAFLD while atezo/bev in patients with viral etiology [132]. ORIENT-32, a phase 2–3 trial compared the use of IBI305, a bevacizumab biosimilar, in combination with sintilimab vs. sorafenib. Patients in the study group achieved a significantly longer OS (HR = 0.57, *p* < 0.0001) as well as PFS (HR = 0.56, *p* < 0.0001) [133]. 

### 4.3. Concluding Remarks

Taken together, the approval of sorafenib was a breakthrough in the treatment of advanced HCC, which paved the way for a number of subsequent trials, therapeutic strategies, and agents, which improved the clinical outcomes of treatment. During the last 15 years, median OS observed in the trials showed a significant improvement, starting at 10.7 months in the landmark SHARP trial, to 22.1 months in the CARES-310 trial or 19.2 months in the IMbrave-150 study. Importantly, the last two trials demonstrated the important clinical efficacy of anti-angiogenic therapeutics combined with immunotherapy. Consequently, many of the described agents have been included in the treatment guidelines of HCC [134,135]. In addition, the described agents have been used in sequential treatment strategies, which resulted in a significantly long survival. For instance, von Felden et al. described 14 patients intolerant or who progressed after sorafenib. These patients were treated with the median of three treatment lines and the median OS from the start of sorafenib was 37.4 months [136]. Moreover, in the previously mentioned RELEVANT study, median OS in the cohort treated sequentially with immunotherapy was 47 months [55]. Nevertheless, it is worth noting that the outcomes of these studies may depend on the baseline characteristics (i.e., BCLC stage B vs. C, Child–Pugh stages, or ECOG performance status, among others). 

Throughout the years, sorafenib has been used as a control arm in evaluating potential novel first-line-treatment agents. Interestingly, median OS and ORRs have been increasing in more recent phase 3 trials. Moreover, shorter treatment duration is being observed as well. In a paper by Brown et al., the authors suggest that these findings may result from improvements in supportive care, anti-viral therapies, or multidisciplinary care coordination. As a result, it may be more difficult to demonstrate superiority to sorafenib in subsequent studies [137]. Multiple clinical studies are currently ongoing in HCC. For instance, the TRIPLET-HCC is a phase 2–3 clinical trial with an estimated completion date in 2026. It will evaluate the addition of ipilimumab (anti-CTLA-4) to an atezolizumab/bevacizumab combination [138]. RACB, a phase 2 single-arm study, will investigate the efficacy of atezo/bev treatment in patients with unresectable HCC, who will undergo a conversion surgical treatment [139]. Nevertheless, new antiangiogenic agents are needed in the treatment of HCC. Aflibercept (a soluble VEGFR decoy) [140], ZLF-095 (VEGFR 1–3 inhibitor) [141], and BD0801 (anti-VEGF monoclonal antibody) [142] are some of the therapeutics that have shown preclinical activity toward HCC. A timeline of trials involving targeted and immunotherapeutics is depicted in Figure 2. Table 1 summarizes described trials involving targeted agents and immunotherapies, while Table 2 lists a few selected phase 3 trials with a future estimated date of completion. 

## 5. Mechanisms of Resistance

### 5.1. Resistance to Targeted Therapies

#### 5.1.1. Sorafenib

Despite being a breakthrough for patients with advanced HCC, the use of TKIs is associated with several challenges that need to be resolved to improve their efficacy, such as sorafenib resistance [154] or dose reductions due to AEs in cabozantinib. Consequently, identification and management of resistance, as well as treatment agents with novel mechanisms of action are greatly needed. In this section, we will discuss some of the possible methods that could overcome resistance to TKI, as well as novel potential targets that have been identified in HCC. 

To begin with, resistance to sorafenib seems to significantly change the gene expression profile in tumor cells. Tovar et al. found 528 differently expressed genes between sorafenib-resistant and -sensitive tumor cells [155]. The resistance can be associated with an activation of other receptors. Among these differently expressed genes, resistant cells showed enrichment of FGF and insulin-like growth factor (IGF) pathways [155]. A combination of sorafenib with AG-1024 (anti-IGF1R) provided significantly higher anti-cancer effects in cells resistant to sorafenib [156]. Moreover, in sorafenib-resistant HCC cells, an activated positive loop between EGFR and Kruppel-like factor 4 (KLF4) has been detected. Inhibition of these molecules could restore the sensitivity to sorafenib [157].

In addition, the yes-associated protein (YAP) pathway has been implicated in the induction of resistance to sorafenib. Studies have demonstrated various mechanisms involving YAP that could be related to the resistance. Firstly, YAP expression is elevated in sorafenib-resistant HCC cells [158]. Furthermore, sorafenib stimulates the anti-apoptotic pathways. Specifically, it enhances YAP to promote survivin, an inhibitor of apoptosis implicated in the pathogenesis of cancers [159]. In an in vivo experiment, a deficiency of YAP improved sorafenib sensitivity [160].

Another mechanism of sorafenib resistance is related to hypoxia, a mechanism that induces hypoxia-inducible factors (HIF), FOXO3a, and autophagy [161]. Under a hypoxic environment, heat shock protein 90 (Hsp90) is positively correlated with HIF-1 and VEGF. The use of ganetespib, a Hsp90 inhibitor, improved the efficacy of sorafenib [162]. However, a phase 1 trial published in 2015 found no clinical efficacy of ganetestib in patients with advanced HCC, who were intolerant or progressed on sorafenib [163]. However, luminespib, another Hsp90 inhibitor, showed antitumor activity in HCC cell lines in in vitro and in vivo experiments [164,165]. The expression of HIF-1 has been found upregulated in sorafenib-resistant HCC compared to sorafenib-sensitive or cases without treatment [166]. Inhibition of HIF by a 32–134D compound and in combination with anti-PD-1 showed significant tumor eradication in mouse models [167]. Anti-cancer activity was also observed by LW6, another HIF-1 inhibitor [168].

#### 5.1.2. Lenvatinib

Similarly, multiple studies have examined the potential resistance mechanisms to lenvatinib. To begin with, bypass signaling may contribute to this process. Few studies have shown an activation of the EGFR as a potential mechanism of lenvatinib resistance [169,170]. In clinical settings, EGFR amplifications detected by circulating tumor DNA (ctDNA) were observed in patients who started to progress while treated with lenvatinib [171]. The use of lenvatinib with an EGFR inhibitor could reverse the resistance in vitro [169]. A few phase 2 studies evaluated the combination of erlotinib (anti-EGFR) with anti-angiogenic agents. An early study by Thomas and colleagues evaluated 40 patients treated with the combination of bevacizumab and erlotinib in patients with advanced HCC who could not be treated with surgical or regional therapies. This included patients who could have been treated previously with one line of systemic treatment, but anti-EGFR or -VGFR therapies were not allowed. The ORR was 25% while the median PFS and OS were 9 months and 15.7 months, respectively [172]. Nevertheless, a few years later a subsequent phase 2 trial investigated the same treatment combination in 10 patients who failed a previous sorafenib treatment. In this cohort, none of the patients achieved an overall response or SD. The median PFS and OS were 1.51 months and 4.37 months, respectively [173]. However, to the best of our knowledge, included patients were not evaluated for the activity of the EGFR as a potential contributing factor to sorafenib resistance. Moreover, this study included a small population with a high percentage of HBV (80%), which could affect the results. In 2016, a phase 2 trial by Kaseb al. was published, which investigated bevacizumab with erlotinib after the failure of sorafenib treatment. In contrast to the previous trial, the sample size was bigger (n = 44) and a much lower percentage of the population carried the B virus (18%). At 16 weeks, the PFS rate was 43%, 4 and 18 patients achieved a PR and SD, respectively, while the median OS was 9.9 months [174]. However, randomized trials evaluating a combination of sorafenib with erlotinib, as well as a comparison between bevacizumab with erlotinib vs. sorafenib in first-line settings did not show statistically improved survival in the erlotinib groups. Small tendencies toward improved response rates were observed [175,176]. Consequently, it remains to be evaluated whether erlotinib could find use in the treatment of a selected population with overexpressed EGFR. Nonetheless, in a paper by Jin and colleagues, the authors found no synergy in EGFR^low^ or EGFR^high^ liver cancer cells between sorafenib and gefitinib (anti-EGFR). In contrast, the authors observed the activation of EGFR by lenvatinib and a synergistic anti-cancer effect of lenvatinib with gefitinib in EGFR^high^ liver cancer cells. Consequently, the NCT04642547 clinical trial aims to investigate the efficacy of lenvatinib with genfitinib in HCC who progressed on lenvatinib [177]. 

In addition, the HGF/c-MET pathway may also mediate the resistance to lenvatinib. In an in vitro experiment, Fu et al. showed that HGF reduced the anti-cancer activity of lenvatinib in cell lines with high expression of c-MET [178]. Silencing c-MET promotes the efficacy of lenvatinib in resistant HCC cell lines [179]. Targeting MET in HCC has been investigated in a few trials. In a phase 2 study, patients with advanced HCC were treated with capmatinib, a MET inhibitor. Interestingly, the study suggested that a better response can be achieved in patients with high MET expression [180]. A randomized phase 2 trial evaluated the use of another MET inhibitor, tepotinib vs. sorafenib in patients with advanced HCC with overexpression of MET. The tepotinib cohort showed significantly longer median PFS and a higher ORR (2.8 vs. 1.4 months, *p* = 0.0229; 10.5% vs. 0%) [181]. Unfortunately, a phase 3 trial evaluating the efficacy of second-line tivantinib vs. placebo in patients with high MET expression did not show improved outcomes in the MET inhibitor group [182]. A promising efficacy has been observed in the phase 1/2 study of foretinib as a first-line treatment in advanced HCC. The ORR was 22.9% and the median OS was 15.7 months [183]. Studies evaluating the efficacy and safety of inhibiting MET in combination or sequentially with lenvatinib remain to be performed. Preclinical anti-cancer activity of agents targeting downstream elements of the RTK signaling pathways has also been demonstrated [184,185,186].

Recently, multiple novel pathways and molecules have been uncovered to play an important role in driving the resistance to lenvatinib. For instance, the upregulation of the c-SRC pathway has been observed in lenvatinib-resistant HCC cells. The use of dasatinib (anti-c-SRC) could promote sensitivity to the multi-TKI [187]. Other molecules implicated in lenvatinib resistance involve cyclin-dependent kinase 6 (CDK6) [188], aldo-keto reductase family 1 member C1 (*AKR1C1*) [189], acylphosphatase 1 (ACYP1) [190], and USP22 [191], among many others.

### 5.2. Resistance to Immunotherapy

Immunotherapeutic agents are currently recommended for the systemic treatment of HCC. However, the ORR in the most recent trials showed a response at approximately 20% [99]. Alterations of the TME are considered to play an important role in driving the resistance to immune agents. Specifically, stimulation of immunosuppressive TME could be related to the ineffectiveness of immunotherapy. HCC tumors resistant to immunotherapy demonstrate reduced CD4+ and CD8+ populations, as well as increased myeloid-derived suppressor cells (MDSCs) and monocytes. Additionally, T cells show upregulated expression of PD-1 and other inhibitory molecules, such as T cell immunoglobulin and mucin domain containing-3 (TIM-3) [192]. Wei et al. showed that zinc finger protein 64 (ZFP64) may be related to the TME phenotype. Tumors derived from mice overexpressing ZFP64 showed a reduced presence of CD4+ and CD8+ T cells, as well as an increased number of M2 anti-inflammatory macrophages. Furthermore, the authors demonstrated that ZFP64 interacts with protein kinase C-alpha (PKC-α) to induce this effect. Importantly, animals treated with anti-PD1 agents with progressed disease showed upregulation of the PKC-α/ZFP64 axis [193]. Moreover, Xiong and collaborators demonstrated that peroxisome proliferator-activated receptor gamma (PPAR-γ) could contribute to the immunosuppressive TME. Furthermore, the authors showed that the combination of PPAR-γ with an anti-PD-L1 inhibitor improved the outcomes. In clinical samples of HCC patients treated with immunotherapy, upregulation of *PPARG* was detected in 40% of patients who did not achieve a durable clinical benefit [192]. In addition, a recent study demonstrated an important role of cancer-associated fibroblasts (CAFs) in modulating the suppressive properties of TME. Specifically, CD36+ CAFs could import oxidized lipids, which stimulated the production of macrophage migration inhibitory factor (MIF), a molecule responsible for tumor accumulation of myeloid-derived suppressor cells. Importantly, the combination of immunotherapy with anti-CD36 increased the anti-cancer efficacy in vivo [194]. In addition, the reduced efficacy of immunotherapy may come from the promotion of other T-cell inhibitory molecules. For instance, through the upregulation of PVRL1, HCC cells could stimulate an inhibitory T-cell immunoreceptor with Ig and ITIM domains (TIGIT). A combined inhibition of PD1 and TIGIT promoted the anti-cancer effects [195]. Overall, the promotion of immunosuppressive TME promotes resistance to immunotherapy. Combinations with agents targeting molecules contributing to the immunosuppressive environment or other inhibitory T-cell molecules could further improve the treatment outcomes. 

### 5.3. Potential Future Targets

As demonstrated above, resistance mechanisms represent an area of active research. Nevertheless, treatment efficacy may be improved by investigating novel targets and targeted therapies. For instance, aurora kinases may represent novel targets in HCC. Aurora kinases A, B, and C are serine/threonine kinases, which take part in the mitosis process. They are involved in the allocation of daughter cells and chromosomal division. Importantly, cancer cells often overexpress the aurora kinases [196]. In HCC, the overexpression of aurora kinases has been observed. Moreover, their high expression has been correlated with poorer survival [197,198]. The use of aurora kinase inhibitors in monotherapy or combined showed important anti-cancer properties in in vitro and in vivo preclinical studies [199,200,201]. Recently, a novel FLT3/AURK inhibitor DBPR114 showed activity against HCC cells, including those resistant to sorafenib. Moreover, the inhibitor was found to target AXL and MET [202].

Another molecule that could become a target in HCC is the tumor growth factor (TGF). TGF-β1 is overexpressed in HCC compared to normal adjacent tissues and its high expression has been associated with poor survival [203]. TGF takes part in a tumor progression process known as epithelial-to-mesenchymal transition (EMT). It can be secreted by the regulatory T cells (Treg) that contribute to the EMT progression [204]. Moreover, TGF can promote the differentiation of Treg cells [205]. Several agents targeting TGF have been developed, including vactosertib, galunisertib, and fresolimumab, among others [206]. In HCC, galunisertib has been evaluated in a phase 2 clinical trial. In this study, patients with advanced HCC were treated with a combination of galunisertib and sorafenib. These patients achieved the median OS of 18.8 months and 4.5% (RECIST)/11.4% (mRECIST) of patients achieved a partial response [207]. Several studies investigated the potential of CDKs to become targets in HCC. CDKs bear several important cellular functions. Depending on the subfamily, these kinases can regulate cell cycle and transcription. Moreover, they can also mediate the activation of other members of the CDK family. CDK inhibitors, such as palbociclib or ribociclib, showed important anti-cancer activity and found use in the treatment of patients with breast cancer. Analyses of clinical HCC tissue samples showed that CDKs can be upregulated in tumor tissues. For instance, Lu and colleagues demonstrated that CDK4 was overexpressed in 73% of cases compared to adjacent normal tissues. Its high expression has been correlated with poor survival [208]. Similar observations were noted in the case of CDK7 [209]. Moreover, higher expression of CDK8 has been found in HCC tissues and it correlated with more advanced disease stages [210]. Taken together, numerous pathways and targets are constantly being investigated for the treatment of HCC. Personalized treatment is a novel and exciting approach to oncologic treatment, which, in the future, may allow patients to be treated with a particular molecular alteration with a specific medicine. Recently, Limousin et al. performed a pilot study of patients with HCC and hepato-cholangiocarcinoma (CCK) refractory to atezo/bev combination. In this study, whole genome/exome/RNA sequencing was performed to guide a subsequent line of treatment and introduce an appropriate targeted agent. For instance, one patient harbored TSC2 alteration, which resulted in the introduction of everolimus (mTOR inhibitor). Consequently, the patient achieved a complete response. The other six patients who received targeted drugs had progressive disease. Nevertheless, 60% of all treated patients (HCC + CCK) were alive after one year [211]. 

In addition, another method to treat HCC in the future could be through targeting cancer stem cells (CSCs). These cells share similarities with regenerative stem cells, but have high tumorigenic potential, and are associated with tumor recurrence and drug resistance [212]. Furthermore, tumor non-stem cells have been suggested to be able to obtain CSC markers, which increases their tumorigenicity [213]. Interestingly, the presence of different subtypes of CSCs could contribute to the intratumor heterogeneity and impact the prognosis of HCC patients [214]. Clinically, the high expression of certain CSC markers has been associated with poor PFS. For instance, Kim et al. analyzed CSC markers in the cohort of HCC patients treated with sorafenib. The authors demonstrated that tissue-based combinational high expression of CD133 and CD90, as well as CD133 with EpCAM, were associated with shorter PFS [215]. Moreover, a greater presence of ICAM-1-positive circulating CSC was observed in HCC patients with microvascular invasion, multiple tumors, as well as with lung metastasis [216]. In addition, CD73 has been found to be associated with HCC stemness, and its overexpression was suppressed, while shRNA-mediated knockdown promoted sensitivity to lenvatinib [217]. Therefore, methods to target and inhibit CSC activity to improve the treatment outcomes are of great interest. In recent years, studies have uncovered various molecules that regulate HCC stemness. For instance, leucine-rich repeat-containing G protein-coupled receptor 5 (lgr5) is a stem cell marker that is detected in a subset of hepatocytes adjacent to the central veins. Lgr5-positive cells can undergo neoplastic transformation and have been suggested to promote the development of HCC [218]. Subsequently, Cao et al. demonstrated the presence of cells enriched with lgr5 in HCC tumors. The population of Lgr5+ cells further increased after treatment with sorafenib or 5-fluorouracil [219]. Lgr5 is associated with the activity of PTEN/AKT and Wnt/β-catenin pathways, the modulation of which changes the population of Lgr5+ cells in hepatic cancer [220]. Few studies examined targeting Lgr5-positive cells in treating cancer and showed promising efficacy of antibody–drug conjugates against Lgr5-high gastrointestinal tumors [221,222]. Nevertheless, future studies should examine the efficacy of agents targeting Lgr5 and whether it could be combined with therapeutics modulating the PTEN/AKT and Wnt/β-catenin pathways in the treatment of HCC.

Furthermore, future studies should evaluate the potential targeting of metabolic pathways to increase the efficacy of HCC treatment. In contrast to normal cells, the malignant ones tend to induce and highly rely on aerobic glycolysis, which is termed the Warburg effect and is associated with the production of lactates [223]. Studies have demonstrated that molecules that enhance aerobic glycolysis, promote tumor progression as well [224,225]. Importantly, lactates cause the acidification of TME, and, as a result, infiltrating immune cells undergo certain adaptations to function under these conditions. For instance, macrophages can increase the expression of carbonic anhydrase XII to enhance survival under acidic conditions. Consequently, these cells can secrete C-C motif chemokine ligand 8 (CCL8) to promote the development of metastasis [226]. Moreover, cells that are present in the TME may also change their metabolic pathways to stimulate tumorigenesis. Chen et al. demonstrated that monocytes derived from HCC tissues show an elevated expression of glycolytic genes, together with a greater activity of lactate dehydrogenase. Interestingly, the promotion of glycolysis was associated with stimulation of PD-L1 expression [227]. In addition, tumor-associated immune cells can enhance glycolysis in HCC cells to further promote tumor progression [228]. Therefore, targeting the glycolysis pathways could find use in the treatment of HCC. Recently, Li and colleagues showed that inhibiting phosphoglycerate kinase 1, an enzyme that is involved in the glycolysis pathway, suppressed the viability of the HCC cells [229]. As a result, targeting glycolysis is an interesting and promising concept in the treatment of HCC that needs further exploration. 

## 6. Conclusions

To conclude, the present review summarizes 15 years of studies of targeted therapy and immunotherapy in HCC. Currently, multiple targeted agents and immunotherapeutics are included in the treatment guidelines for patients with advanced HCC. Based on the reviewed literature, several points can be made. Firstly, more recent trials tend to investigate the combinations of targeted agents with immunotherapy, and these combinations show important clinical benefits. Secondly, sequential treatment with multiple drugs shows exceptional survival in advanced HCC. Interestingly, despite novel treatment agents, phase 3 trials show that patients included in the sorafenib control arms achieve better outcomes than in the landmark SHARP trial. With abundant studies demonstrating the promising efficacy of multiple molecules and combinations, future studies should focus on evaluating markers that could drive the selection of treatment methods in specific populations. Thus, we could identify patients more likely to benefit from a particular therapy and improve treatment outcomes. Furthermore, understanding the resistance mechanisms is necessary to improve the clinical efficacy of approved agents. Similarly, new therapeutics in the treatment of HCC are greatly needed, but current evidence shows that translating preclinical anti-cancer activity into clinical outcomes is challenging and many preclinical observations do not translate into clinical results. 

## Figures and Tables

**Figure 1 ijms-25-01456-f001:**
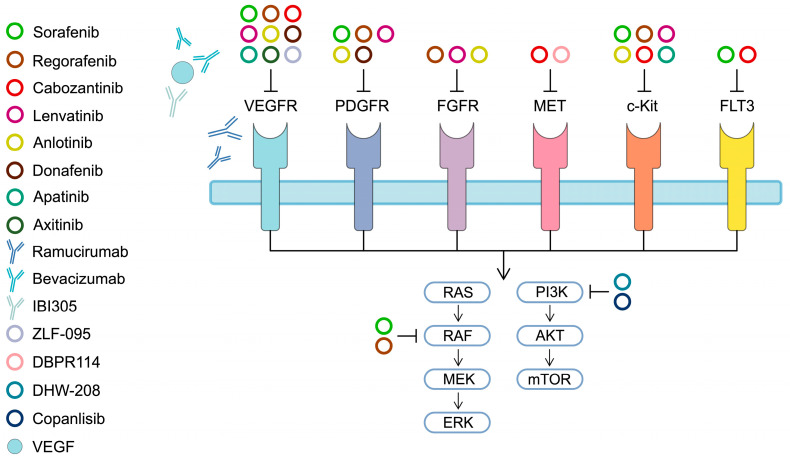
A simplified representation of mechanism of action of targeted therapeutics. AKT—protein kinase B; ERK—extracellular signal-regulated kinase; FGFR—fibroblast growth factor receptor; FLT-3—FMS-like tyrosine kinase 3; MEK—mitogen-activated protein kinase; mTOR—mammalian target of rapamycin; PDGFR—platelet-derived growth factor receptor; PI3K—phosphatidylinositol 3-kinase; RAF—rapidly accelerated fibrosarcoma; RAS—rat sarcoma virus; VEGFR—vascular endothelial growth factor receptor.

**Figure 2 ijms-25-01456-f002:**
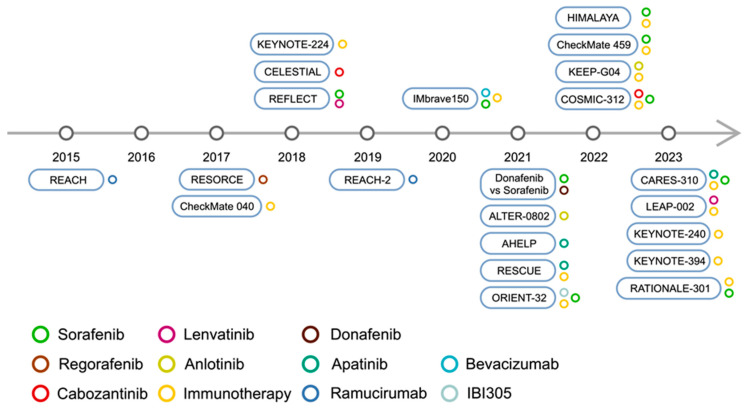
A timeline of selected phase 2/3 HCC trials published after 2015.

**Table 1 ijms-25-01456-t001:** Summary of selected phase 2 and 3 clinical trials investigating targeted agents and immunotherapies in hepatocellular carcinoma.

Investigated Therapy	Publication Year	Phase	Line of Systemic Treatment	Arms	Number of Participants	Median Overall Survival or Progression-Free Survival (Months)	Response (%)	Reference
Sorafenib vs. Placebo (SHARP, NCT00105443)	2008	III	First	Sorafenib	299	OS: 10.7	PR: 2 SD: 71	[21]
				Placebo	303	OS: 7.9	PR: 1 SD: 67	
Sorafenib vs. Placebo (Asia Pacific, NCT00492752)	2009	III	First	Sorafenib	150	OS: 6.5	PR: 3.3 SD: 54	[22]
				Placebo	76	OS: 4.2	PR: 1.3 SD: 27.6	
Sorafenib plus Doxorubicin vs. Doxorubicin plus Placebo (NCT00108953)	2010	II	First	Sorafenib + Doxorubicin	47	OS: 13.7 PFS: 6	PR: 4	[23]
				Doxorubicin + Placebo	49	OS: 6.5 PFS: 2.7	CR: 2	
Sorafenib plus Doxorubicin vs. Sorafenib alone (NCT01015833)	2019	III	First	Sorafenib + doxorubicin	180	OS: 9.3 PFS: 4	CR: 0.7 PR: 9.3	[24]
				Sorafenib	176	OS: 9.4 PFS: 3.7	PR: 5.4	
Sorafenib plus doxorubicin (NCT01840592)	2020	II	Second	Sorafenib + Doxorubicin	30	OS: 8.6 PFS: 3.9	PR: 10.7 SD: 60.7	[25]
Sorafenib plus low-dose cisplatin and fluorouracil HAIC vs. Sorafenib alone (SILIUS, NCT01214343)	2018	III	-	Sorafenib + HAIC	102	OS: 11.8 PFS: 4.8	CR: 8 PR: 28 SD: 28	[26]
				Sorafenib	103	OS: 11.5 PFS: 3.5	CR: 2 PR: 16 SD: 55	
Sorafenib Plus Oxaliplatin, 5-fluorouracil HAIC vs. Sorafenib alone (NCT03009461)	2022	II	First	Sorafenib plus HAIC	32	OS: 16.3 PFS: 9	ORR: 41 [RECIST] ORR: 50 [mRECIST]	[28]
				Sorafenib	32	OS: 6.5 PFS: 2.5	ORR: 3	
Sorafenib plus Bavituximab (NCT01264705)	2019	II	First	Sorafenib + Bavituximab	38	OS: 6.1 PFS: 4.8	ORR: 5	[29]
Sorafenib Plus vitamin K vs. Sorafenib alone	2020	II	First	Sorafenib + vitamin K	22	OS: 12 PFS: 4.9	ORR: 27.3	[30]
				Sorafenib	22	OS: 11.5 PFS: 2.7	ORR: 4.5	
Regorafenib (NCT01003015)	2013	II	Second	Regorafenib	36	OS: 13.8	ORR: 3 DCR: 72	[42]
Regorafenib vs. placebo (RESORCE, NCT01774344)	2017	III	Second	Regorafenib	379	OS: 10.6 PFS: 3.1	ORR: 11 DCR: 65	[43]
				Placebo	194	OS: 7.8 PFS: 1.5	ORR: 4 DCR: 36	
Cabozantinib vs. placebo (CELESTIAL, NCT01908426)	2018	III	Patients could have received ≤ 2 prior systemic treatments	Cabozantinib	470	OS: 10.2 PFS: 5.2	ORR: 4 Disease control (PR + SD): 64	[47]
				Placebo	237	OS: 8 PFS: 1.9	ORR: <1 Disease control (PR + SD): 33	
Cabozantinib plus atezolizumab vs. sorafenib (COSMIC-312, NCT03755791)	2022	III	First	Cabozantinib + Atezolizumab	PFS ITT population: 250	PFS: 6.8	ORR: 13	[113]
				Sorafenib	PFS ITT population: 122	PFS: 4.2	ORR: 5	
				Cabozantinib + Atezolizumab	ITT: population: 432	OS: 15.4	ORR: 11	
				Sorafenib	ITT population: 217	OS: 15.5	ORR: 4	
Lenvatinib vs. Sorafenib (REFLECT, NCT01761266)	2018	III	First	Lenvatinib	478	OS: 13.6 PFS: 7.4	ORR investigator: 24.1 ORR masked independent imaging review mRECIST: 40.6	[53]
				Sorafenib	476	OS: 12.3 PFS: 3.7	ORR investigator: 9.2 ORR masked independent imaging review mRECIST: 12.4	
Anlotinib plus toripalimab	2023	II	First	Anlotinib + Toripalimab	31	OS: 18.2 PFS: 11	ORR: 29 (irRECIST/RECIST) ORR: 32.3 (mRECIST)	[118]
Anlotinib plus Sintilimab (KEEP-G04, NCT04052152)	2022	II	First	Anlotinib + Sintilimab	20	OS: Not reached (95% CI 16.3 to not reached) PFS: 12.2	Confirmed only: ORR (RECIST): 35 ORR (mRECIST): 55	[119]
Axitinib	2015	II	Second	Axitinib	30	OS: 7.1 PFS: 3.6	RECIST 1.1 PR: 11.5 SD: 77 mRECIST 1.1 CR: 3.8 PR: 23.1 SD: 53.8	[65]
Axitinib (NCT01273662)	2020	II	Second	Axitinib	45	OS: 10.1 PFS: 2.2	Response rate: 6.7	[64]
Axitinib vs. placebo (NCT01210495)	2015	II	Second	Axitinib	134	OS: 12.7 PFS: 3.6	ORR: 9.7	[66]
				Placebo	68	OS: 9.7 PFS: 1.9	ORR: 2.9	
Donafenib vs. Sorafenib	2021	II–III	First	Donafenib	328	OS: 12.1 PFS: 3.7	ORR: 4.6	[69]
				Sorafenib	331	OS: 10.3 PFS: 3.6	ORR: 2.7	
Apatinib vs. Placebo (AHELP, NCT02329860)	2021	III	Second and later	Apatinib	261	OS: 8.7 PFS: 4.5	ORR: 11	[73]
				Placebo	132	OS: 6.8 PFS: 1.9	ORR: 2	
Apatinib plus camrelizumab (RESCUE)	2021	II	First	Apatinib + Camrelizumab	70	PFS: 5.7	ORR: 34.3	[121]
			Second	Apatinib + Camrelizumab	120	PFS: 5.5	ORR: 22.5	
Apatinib plus camrelizumab (CARES-310, NCT03764293)	2023	III	First	Apatinib + Camrelizumab	272	OS: 22.1 PFS: 5.6	ORR: 25	[122]
				Sorafenib	271	OS: 15.2 PFS: 3.7	ORR: 6	
Apatinib plus sintilimab plus capecitabine (NCT04411706)	2022	II	First	Apatinib + Sintilimab + Capecitabine	46	OS: not reached PFS: 9	ORR: 50	[123]
Ramucirumab vs. placebo (REACH, NCT01140347)	2015	III	Second	Ramucirumab	283	OS: 9.2 PFS: 2.8	ORR: 7	[75]
				Placebo	282	OS: 7.6 PFS: 2.1	ORR: <1	
Ramucirumab vs. placebo (REACH-2, NCT02435433)	2019	III	Second	Ramucirumab	197	OS: 8.5 PFS: 2.8	ORR: 5	[76]
				Placebo	95	OS: 7.3 PFS: 1.6	ORR: 1	
Atezolizumab plus bevacizumab vs. sorafenib (IMBrave150, NCT03434379)	2020	III	First	Atezolizumab + Bevacizumab	336	OS: could not be evaluated PFS: 6.8	RECIST ORR: 27.3 mRECIST ORR: 33.2	[126]
				Sorafenib	165	OS: 13.2 PFS: 4.3	RECIST ORR: 11.9 mRECIST ORR: 13.3	
Updated efficacy and safety: Atezolizumab plus bevacizumab vs. sorafenib	2022			Atezolizumab + Bevacizumab	336	OS: 19.2 PFS: 6.9	ORR: 30	[127]
				Sorafenib	165	OS: 13.4 PFS: 4.3	ORR: 11	
Sintilimab plus IBI305 vs. sorafenib (ORIENT-32, NCT03794440)	2021	II–III	First	Sintilimab + IBI305	380	OS: not reached PFS: 4.6	RECIST ORR: 21 mRECIST ORR: 24	[133]
				Sorafenib	191	OS: 10.4 PFS: 2.8	RECIST ORR: 4 mRECIST ORR: 8	
Nivolumab (CheckMate 040, NCT01658878)	2017	II	Previous systemic treatment in 74% of patients	Nivolumab	214	OS: not reached PFS: 4	ORR: 20	[79]
Pembrolizumab (KEYNOTE-224, NCT02702414)	2018	II	Second	Pembrolizumab	104	OS: 12.9 PFS: 4.9	Objective response: 17	[85]
Pembrolizumab vs. Placebo (KEYNOTE-240, NCT02702401)	2020	III	Second	Pembrolizumab	278	OS: 13.9 PFS: 3.0	ORR: 18.3	[87]
				Placebo	135	OS: 10.6 PFS: 2.8	ORR: 4.4	
Pembrolizumab vs. Placebo (KEYNOTE-394, NCT03062358)	2023	III	Second	Pembrolizumab	300	OS: 14.6 PFS: 2.6	ORR: 13.7	[89]
				Placebo	153	OS: 13.0 PFS: 2.3	ORR: 1.3	
Pembrolizumab (KEYNOTE-224cohort 2, NCT02702414)	2022	II	First	Pembrolizumab	51	OS: 17 PFS: 4	ORR: 16	[90]
Tislelizumab (RATIONALE-208, NCT03419897)	2022	II	≥1 previous systemic treatment	Tislelizumab	249	OS: 13.2 PFS: 2.7	ORR: 13	[94]
Tislelizumab vs. Sorafenib (RATIONALE-301, NCT03412773)	2023	III	First	Tislelizumab	338	OS: 15.9 PFS: 2.1	ORR: 14.3	[95]
				Sorafenib	324	OS: 14.1 PFS: 3.4	ORR: 5.4	
Tremelimumab + Durvalumab vs. Durvalumab vs. Sorafenib (HIMALAYA, NCT03298451)	2022	III	First	Tremelimumab + Durvalumab	393	OS: 16.43 PFS: 3.78	ORR: 20.1	[99]
				Durvalumab	389	OS: 16.56 PFS: 3.65	ORR: 17	
				Sorafenib	389	OS: 13.77 PFS: 4.07	ORR: 5.1	

DCR—disease control rate (CR + PR + SD); HAIC—hepatic arterial infusion chemotherapy; irRECIST—immune-related Response evaluation criteria in solid tumors; mRECIST—Modified response evaluation criteria in solid tumors; ORR—objective response rate; OS—Overall survival; PFS—progression-free survival; RECIST—Response evaluation criteria in solid tumors.

**Table 2 ijms-25-01456-t002:** A list of selected phase 3 clinical trials investigating targeted agents and immunotherapies in patients with advanced/unresectable HCC with estimated completion date in the future with statuses “Recruiting” and “Active, not recruiting” registered at the https://www.clinicaltrials.gov.

Registration Number	Investigated Therapy	Arms	Estimated Study Completion Year	Reference
NCT04246177, LEAP-012	Lenvatinib with Pembrolizumab and TACE	Lenvatinib Pembrolizumab TACE	2029	[143]
		Oral placebo, intravenous placebo, TACE		
NCT04039607, CheckMate 9DW	Nivolumab with Ipilimumab	Nivolumab Pembrolizumab	2026	[144]
		Sorafenib/Lenvatinib		
NCT04723004	Toripalimab with Bevacizumab	Toripalimab Bevacizumab	2024	[145]
		Sorafenib		
NCT04523493	Toripalimab with Lenvatinib	Toripalimab Lenvitinib	2026	[146]
		Placebo Lenvatinib		
NCT05883644, SIERRA	Durvalumab with Tremelimumab	Durvalumab Tremelimumab	2025	[147]
NCT04194775	Nofazinlimab	Nofazinlimab	2025	[148]
NCT05608200	Lanvatinib with Sintilimab with TACE	Lenvatinib Sintilimab TACE	2026	[149]
		Lenvatinib TACE		
NCT05985798	Sintilimab with Bevacizumab with TACE	Sintilimab Bevacizumab TACE	2027	[150]
		Lenvatinib TACE		
NCT05904886, IMbrave152, SKYSCRAPER-14	Atezolizumab with Bevacizumab with Tiragolumab	Atezolizumab Bevacizumab Tiragolumab	2026	[151]
		Atezolizumab Bevacizumab Placebo		
NCT06172205	FOLFOX with Camrelizumab with Apatinib	Intravenous FOLFOX Camrelizumab Apatinib	2027	[152]
		HAIC-FOLFOX Camrelizumab Apatinib		
NCT04560894	SCT-I10A with SCT510	SCT-I10A SCT510	2024	[153]
		Sorafenib		

HAIC—hepatic arterial infusion chemotherapy; TACE—transarterial chemoembolization.

## Data Availability

Not applicable.

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
