# Peer review of "New Opportunities in the Systemic Treatment of Hepatocellular Carcinoma—Today and Tomorrow"

_ijms, 2024, doi:10.3390/ijms25031456_

Round 1

Reviewer 1 Report

Comments and Suggestions for Authors

The document is a comprehensive review titled "New Possibilities in the Systemic Treatment of Hepatocellular Carcinoma—Today and Tomorrow." It focuses on the most common primary liver cancer, Hepatocellular carcinoma (HCC), discussing its major risk factors, such as liver cirrhosis, hepatitis B and C, and nonalcoholic fatty liver disease. The review extensively covers the evolution of systemic treatments for HCC, especially in the last fifteen years, highlighting the expansion of available therapeutics and improved treatment outcomes. It also delves into the need for novel combinations and drugs due to varying patient responses. The paper reviews trials investigating the safety and efficacy of targeted therapeutics and immunotherapies since the introduction of sorafenib, a key drug in HCC treatment. Additionally, it examines current evidence regarding resistance mechanisms and potential novel targets in treating advanced HCC.

Your article is excellent, but my suggestion is to add a paragraph to your article that represents the future direction. Of course, the control of liver cancer stem cells is also a necessary direction.

In mice, Lgr5 and its progeny cells appear only under certain injury conditions (Barker N, Gastroenterology, 2010). Recent studies indicate that Lgr5 and its progeny cells emerge in liver cancer, and are associated with the PTEN-Wnt/β-catenin pathway (Ang CH, PNAS, 2019) (J Han, IJBS, 2021) (J He, BBRC, 2023). Therefore, I encourage you to seriously consider the relationship between Lgr5 and your review, as it may add some interesting elements. Your single-cell data can create connections for basic research. Lgr5 serves as an excellent bridge.

For example:

Recent research shows that drug delivery systems using antibodies targeting Lgr5+ cells have proven effective in cancer treatment. Two antibody-drug conjugates (ADCs) targeting LGR5, by attaching monomethyl auristatin E to anti-LGR5 antibodies, successfully eradicated gastrointestinal tumors and CSCs with high LGR5 levels. Additionally, two other ADCs targeting Lgr5+ cells in colon cancer showed significant anti-tumor effects by reducing tumor size and proliferation. However, cancer treatment often faces resistance. Combining small molecule drugs that inhibit CSC pathways with ADCs is anticipated to be an effective strategy. Thus, using ADCs together with drugs targeting PTEN/AKT and Wnt/β-catenin signaling pathways could enhance the potential of Lgr5+ cells in liver cancer (HCC) treatment, limiting their expansion(X. Gong, Mol. Cancer Therapeut.,2016) (J He, BBRC, 2023) (M.R. Junttila, Sci. Transl. Med. ,2015).

.

Comments on the Quality of English Language

Minor editing of English language required

Author Response

The document is a comprehensive review titled "New Possibilities in the Systemic Treatment of Hepatocellular Carcinoma—Today and Tomorrow." It focuses on the most common primary liver cancer, Hepatocellular carcinoma (HCC), discussing its major risk factors, such as liver cirrhosis, hepatitis B and C, and nonalcoholic fatty liver disease. The review extensively covers the evolution of systemic treatments for HCC, especially in the last fifteen years, highlighting the expansion of available therapeutics and improved treatment outcomes. It also delves into the need for novel combinations and drugs due to varying patient responses. The paper reviews trials investigating the safety and efficacy of targeted therapeutics and immunotherapies since the introduction of sorafenib, a key drug in HCC treatment. Additionally, it examines current evidence regarding resistance mechanisms and potential novel targets in treating advanced HCC.

Your article is excellent, but my suggestion is to add a paragraph to your article that represents the future direction. Of course, the control of liver cancer stem cells is also a necessary direction.

In mice, Lgr5 and its progeny cells appear only under certain injury conditions (Barker N, Gastroenterology, 2010). Recent studies indicate that Lgr5 and its progeny cells emerge in liver cancer, and are associated with the PTEN-Wnt/β-catenin pathway (Ang CH, PNAS, 2019) (J Han, IJBS, 2021) (J He, BBRC, 2023). Therefore, I encourage you to seriously consider the relationship between Lgr5 and your review, as it may add some interesting elements. Your single-cell data can create connections for basic research. Lgr5 serves as an excellent bridge.

For example:

Recent research shows that drug delivery systems using antibodies targeting Lgr5+ cells have proven effective in cancer treatment. Two antibody-drug conjugates (ADCs) targeting LGR5, by attaching monomethyl auristatin E to anti-LGR5 antibodies, successfully eradicated gastrointestinal tumors and CSCs with high LGR5 levels. Additionally, two other ADCs targeting Lgr5+ cells in colon cancer showed significant anti-tumor effects by reducing tumor size and proliferation. However, cancer treatment often faces resistance. Combining small molecule drugs that inhibit CSC pathways with ADCs is anticipated to be an effective strategy. Thus, using ADCs together with drugs targeting PTEN/AKT and Wnt/β-catenin signaling pathways could enhance the potential of Lgr5+ cells in liver cancer (HCC) treatment, limiting their expansion(X. Gong, Mol. Cancer Therapeut.,2016) (J He, BBRC, 2023) (M.R. Junttila, Sci. Transl. Med. ,2015).

Dear Reviewer,

Thank you very much for your time and review of our manuscript. We highly appreciate your kind words. We strongly agree that the control of cancer stem cells (CSCs) is a necessary direction that future studies should follow. Consequently, we have introduced a novel paragraph in the “5.3 Potential Future Targets” section. As suggested, we discussed the promising role of Lgr5 as a possible future target, and we have cited most of papers mentioned by the Reviewer. We strongly believe that included fragment about CSCs improves the quality of our manuscript.

Kind regards,

Authors

Reviewer 2 Report

Comments and Suggestions for Authors

The manuscript by Becht R et al is a comprehensive review of modern approaches in the therapy of a difficult to treat malignant pathology - hepatocellular carcinoma.

The review will be important for oncologists and researchers in the field of cancer research. The authors analyzed a large amount of published data, ongoing clinical trials to present the current view on the progress and problems in the treatment of hepatocellular carcinoma.

There are some minor issues with the submitted manuscript.

1) Table 2. Include immunotherapeutic drugs, but it precedes the chapter on immunotherapy. Perhaps it would be wise to introduce the separate chapter on combinatorial treatment with targeted drugs and immunotherapeutic drugs.

2) Readers will benefit from the internet links included in the table on clinical trials.

3) There are many papers in Pubmed about the new strategy to treat hepatocellular carcinoma with CAR-T cells. The chapter on CAR-T therapy should be added to the immunotherapy section, or at least this novel approach should be discussed in the future perspectives section

Author Response

Dear Reviewer,

Thank you very much for reviewing our manuscript. We greatly appreciate your kind words and insightful comments. Below please find responses to each comment.

The manuscript by Becht R et al is a comprehensive review of modern approaches in the therapy of a difficult to treat malignant pathology - hepatocellular carcinoma. 

The review will be important for oncologists and researchers in the field of cancer research. The authors analyzed a large amount of published data, ongoing clinical trials to present the current view on the progress and problems in the treatment of hepatocellular carcinoma.

There are some minor issues with the submitted manuscript.

  • Table 2. Include immunotherapeutic drugs, but it precedes the chapter on immunotherapy. Perhaps it would be wise to introduce the separate chapter on combinatorial treatment with targeted drugs and immunotherapeutic drugs.

We agree that the separate paragraph about combination treatment might improve the readability of the article. Therefore, as suggested, we have introduced a novel paragraph after the sections discussing immunotherapy. Moreover, we have moved the table after this fragment so an appropriate introduction to immunotherapeutics is included before the appearance of the table.

  • Readers will benefit from the internet links included in the table on clinical trials.

We have included references to each clinical trial summarized in the table. Moreover, we have changed the table to improve readability and included few more studies.

  • There are many papers in Pubmed about the new strategy to treat hepatocellular carcinoma with CAR-T cells. The chapter on CAR-T therapy should be added to the immunotherapy section, or at least this novel approach should be discussed in the future perspectives section.

As suggested, paragraph about CAR-T therapy was added.

Once again, we thank the Reviewer for your time and insightful comments. We hope that our corrections improve the quality of our manuscript.

Kind regards,

Authors

Reviewer 3 Report

Comments and Suggestions for Authors

In their review titled “New Possibilities in the Systemic Treatment of Hepatocellular Carcinoma (HCC)—Today and Tomorrow”, the authors summarize the safety and efficacy of targeted therapeutics and immunotherapies in the treatment of advanced HCC. They also discuss the resistance mechanisms and potential novel targets in the treatment of advanced HCC. The approach to review the overall clinical research which led to approval in the current treatment of HCC is so exhaustive and reference selection very difficult. There are several reviews already published on ‘advances in systemic therapies in HCC’, so I suggest you focus on the newest facts. The current review seems to have too much explanation and overlap. To improve this review, I suggest some points should be addressed.

-Some abbreviations appear without explanation of the full name. The expression methods of some abbreviations are not unified (e.g., irAEs vs. immune-mediated AEs).

- A recent similar review highlighting the ‘perspective on the future of advanced HCC therapies’ (Zhang et al. Biomarker Research (2022) 10:3) should be cited. It would be good to introduce new future HCC treatments such as CAR-T cell therapy.

- Please include explanations of abbreviations included in the figure or table below all figure (Figure 1 & 2) and tables (Table 1 & 2).

- Since Table 1 and Table 3 have overlapping content, it would be better to combine them. Additionally, it would be good to indicate the clinical trials’ number of important clinical studies along with the study name (e.g., SHARP, NCT00105443) in the tables.

- “2.11. Targeted Therapies – Concluding Remarks and Future Trials”: This part would be better to combine in the conclusion (5. Conclusions).

- “4. Mechanisms of Resistance”: It seems to overlap with the previous chapters of ‘2. Targeted Therapy in HCC’ and ‘3. Immunotherapy in HCC’, so it would be better to combine them together at the front.

- Overall, research on intermediate-stage HCC and research on advanced HCC are mixed, so it would be better to distinguish between them. [refer to ‘Llovet et al. NATURE REVIEWS | DISEASE PRIMERS | Article citation I­D: (2021) 7:6’]

Comments on the Quality of English Language

There are some minor typographical errors that should be corrected (points, spaces, grammar-related, mellitus in cursive …). 

Author Response

Dear Reviewer,

Thank you for reviewing our manuscript and providing us with insightful comments. Below please find responses to each suggestion.

In their review titled “New Possibilities in the Systemic Treatment of Hepatocellular Carcinoma (HCC)—Today and Tomorrow”, the authors summarize the safety and efficacy of targeted therapeutics and immunotherapies in the treatment of advanced HCC. They also discuss the resistance mechanisms and potential novel targets in the treatment of advanced HCC. The approach to review the overall clinical research which led to approval in the current treatment of HCC is so exhaustive and reference selection very difficult. There are several reviews already published on ‘advances in systemic therapies in HCC’, so I suggest you focus on the newest facts. The current review seems to have too much explanation and overlap. To improve this review, I suggest some points should be addressed.

We have added and discussed several studies that were most recently published to improve the relevance of our manuscript.

-Some abbreviations appear without explanation of the full name. The expression methods of some abbreviations are not unified (e.g., irAEs vs. immune-mediated AEs). 

It has been corrected.

- A recent similar review highlighting the ‘perspective on the future of advanced HCC therapies’ (Zhang et al. Biomarker Research (2022) 10:3) should be cited. It would be good to introduce new future HCC treatments such as CAR-T cell therapy. 

The abovementioned review was cited and a fragment about CAR-T cells was introduced.

- Please include explanations of abbreviations included in the figure or table below all figure (Figure 1 & 2) and tables (Table 1 & 2). 

Abbreviations have been explained under Figure 1 and Tables 1-2. Figure 2 depicts timeline with names of clinical trials that are not abbreviations.

- Since Table 1 and Table 3 have overlapping content, it would be better to combine them. Additionally, it would be good to indicate the clinical trials’ number of important clinical studies along with the study name (e.g., SHARP, NCT00105443) in the tables.

We have decided to remove the previous Table 1, as not only it had overlapping content with Table 3, but also with Figure 1. Furthermore, we have introduced the registration numbers along with study names in the the current Table1. 

- “2.11. Targeted Therapies – Concluding Remarks and Future Trials”: This part would be better to combine in the conclusion (5. Conclusions).

We agree with the Reviewer that this fragment could fit in the conclusions. Nevertheless, our article is composed of the two main parts – efficacy and safety of targeted agents, immunotherapy and combination strategies, and the part about resistance mechanisms and novel targets. Our manuscript underwent minor structure re-organization, as fragments about combinations of targeted therapies and immunotherapies were grouped and included after the paragraphs on immunotherapy. Therefore, to emphasize the high value of these drugs, we decided to insert the “Concluding remarks and future trials” after the first major part of our manuscript. Consequently, the final “Conclusions” can be more coherent and refer to both clinical aspect (drug efficacy), as well as to preclinical fragments (resistance mechanisms and future targets). 

- “4. Mechanisms of Resistance”: It seems to overlap with the previous chapters of ‘2. Targeted Therapy in HCC’ and ‘3. Immunotherapy in HCC’, so it would be better to combine them together at the front.

As mentioned above, our primary idea for this manuscript was to create two different parts. The first one mainly focuses on the clinical aspects, and the second one discusses resistance mechanisms and potential future targets, thus focusing on the basic and preclinical research. Since the first part mainly cites clinical trials and so called “real-world” data, we strongly believe that the current division improves readability of our manuscript. We strongly believe that future research and discoveries will demonstrate that resistance mechanisms can be identified, which would allow to introduce the next targeted treatments to overcome them. Nevertheless, since these studies are on an early level, we wanted to demonstrate various concepts of resistance mechanisms to clinicians, and thus we implemented these pieces of information in a separate paragraph.

- Overall, research on intermediate-stage HCC and research on advanced HCC are mixed, so it would be better to distinguish between them. [refer to ‘Llovet et al. NATURE REVIEWS | DISEASE PRIMERS | Article citation I­D: (2021) 7:6’]

The aim of our paper was to summarize systemic treatments used today and review current evidence on the novel agents that could be useful in the future. First, systemic treatment is indicated in certain patients with BCLC stage B, as well as in patients with BCLC stage C. Consequently, clinical trials and other clinical studies include patients at these two stages in their study groups. For instance, in the RESORCE trial [Bruix et al. Lancet 2017], there were 14% and 86% of patients with BCLC stages B and C, respectively. In the REACH-2 trial [Zhu et al., Lancet Oncol 2019], B and C stages were in 17% and 83% of patients, respectively. Similarly, among participants in the study group in the AHELP trial [Qin et al., Lancet Gastroenterol Hepatol 2021], 11% of patients had BCLC B stage HCC. Consequently, changing the manuscript to divide BCLC stages B and C would only be possible in case every one of these clinical trials published detailed subgroup analysis demonstrating the safety and efficacy of investigated drugs depending on these stages. Furthermore, it would substantially change the structure of the manuscript. The concept of the review involved the use of systemic treatments which are used in patients with BCLC stage B as well. Nevertheless, we highly agree that the efficacy of these drugs may depend on the clinical stage. Consequently, we have included this sentence in the manuscript:

Nevertheless, it is worth noting that the outcomes of these studies may depend on the baseline characteristics (i.e., BCLC stage B vs C, Child Pugh stages, or ECOG performance status, among others).

In the future papers, we will describe the efficacy of systemic treatments with more focus on the clinical stage of the disease.

To conclude, we thank the Reviewer once again for reviewing our manuscript and for the insightful comments. We hope that implemented corrections improved the quality and readability of the manuscript. Here, we described our view of the paper, nevertheless, we are eager of cooperate if any more corrections are needed.

Kind regards,

Authors

Round 2

Reviewer 1 Report

Comments and Suggestions for Authors

It can publish. I agree to publish.

Author Response

Thank You Very much for all of your help :)

Best Regards,

Authors